# Radiologic Imaging of the In Vivo Position of the New Supraglottic Airway Device Spritztube^®^ in an Adult Patient—A Case Report

**DOI:** 10.3390/diagnostics12122907

**Published:** 2022-11-22

**Authors:** Silvia De Rosa, Massimiliano Sorbello, Alessandro Rigobello, Lucia Cattin, Giuseppe Iannucci, Paolo Gennaro, Vinicio Danzi, Stefano Checcacci Carboni

**Affiliations:** 1Department of Anesthesiology and Intensive Care Medicine, AULSS 8 Berica, San Bortolo Hospital, Viale Rodolfi 37, 36100 Vicenza, Italy; 2Anesthesia and Intensive Care, AOU Policlinico Vittorio Emanuele, 95123 Catania, Italy; 3Department of Neuroradiology, San Bortolo Hospital, 36100 Vicenza, Italy

**Keywords:** airway management, difficult airway, supraglottic airway device, Spritztube

## Abstract

Spritztube^®^ is a new supraglottic airway device that allows either extraglottic ventilation or orotracheal intubation with the same device. The aim of the present report is to provide the first radiologic images of the Spritztube in situ in a living human and to assess the depth of insertion and its anatomical relationships in vivo. We present the case of a 55-year-old man who was admitted to our centre to perform an interventional neuroradiological procedure. We obtained and analysed radiologic images of the head and neck of an adult patient to ascertain the position of the cuffs of the Spritztube relative to different anatomic structures. The insertion and depth of the device, correct tip positioning, effects of the distal and proximal cuffs on adjacent soft tissues, and the position of the pharyngeal cuff from the cranial to the hyoid bone were evaluated. Our report shows that Spritztube could be helpful in maintaining an adequate airway during radiologic procedures.

## 1. Background

The number of supraglottic airways (extraglottic devices) has progressively increased in recent years [1]. As a general principle, these devices allow access to the distal airway with a pharyngeal, laryngeal, or pharyngolaryngeal seal, enabling oxygenation and ventilation by creating an adequate seal.

Moreover, some extraglottic devices allow the passage of a tracheal tube, which increases the success and safety of fibreoptic bronchoscopy. A re-usable extraglottic, the Spritztube [^®^Med Europe s.r.l., San Pietro in Casale, Italy], has been designed, developed, and tested in our centre. It consists of a silicone-based tracheal tube with two low-pressure cuffs. The proximal cuff is designed to seal the oropharynx above the epiglottis and the distal cuff is designed to seal the upper oesophagus (Figure 1 and Figure 2). Cuffs are inflated through independent tubing. The Spritztube comes with a long stylet designed to stiffen the device and allow passage through the oropharyngeal space. The positioning of this device requires the aid of this dedicated stylet, designed to align the two cuffs while stiffening the Spritztube so as to allow passage through the oropharynx down to the hypopharynx. The Spritztube can be used for extraglottic ventilation, for tracheal intubation with the direct and indirect laryngoscopes, or for fibreoptic-assisted tracheal intubation [2].

The Spritztube is designed for blind insertion. One holds it as a pen and the device is inserted into the oral cavity following the soft palate, stopping introduction once the black marker on the Spritztube (teeth mark) reaches the upper incisors (Figure 1).

Precisely, a marker on the device, positioned at the incisors, indicates the adequate depth of insertion so that the distal cuff is above the oesophagus and the proximal cuff is above the laryngeal opening (Figure 2 and Figure 3).

Once in position, the proximal cuff is inflated up to 60 cc, until no air leak is perceived. The distal cuff, inside the upper oesophageal sphincter, is inflated using approximately 20 cc of air, and then the stylet is removed. 

This position allows the Spritztube to function as a extraglottic device. If endotracheal intubation is needed, the proximal cuff is deflated, and the tube is addressed into the trachea without repositioning the stylet, under direct or indirect laryngoscope, flexible endoscope, or video-stylet view.

Once intubation has been performed, the distal cuff can either be deflated or left in position to act as an oesophageal obturator [2] (Figure 4). As a normal endotracheal tube, average distances such as the 20/22 or 21/23 (cm, female/male) rule have been simply applied.

The Spritztube is available in unique and adult sizes, and it is recently available on the market (given the recent certification mark for conformity with health, safety, and environmental protection standards for products sold within the European market). It was provided for our case report in order to elucidate how it might allow for airway control and whether the two cuffs distort the pharyngeal anatomy. 

This is the first report that provides the first images of the Spritztube device in situ in a living human and assesses the depth of insertion and its positioning in vivo. The patient was part of a clinical trial published previously. We obtained and analysed radiologic images of the head and neck of an adult patient under general anaesthesia to ascertain the position of the cuffs of the Spritztube relative to anatomic structures. Written informed consent was obtained from the patient to publish this case report.

## 2. Case Presentation

A patient enrolled in a first-in-human trial of the Spritztube. This trial compared Spritztube tracheal cannula with the Laryngeal Mask Airway Supreme in anaesthetised adult patients, showing the effectiveness of the Spritztube in maintaining the airway with all patients being successfully ventilated without difficulty. The success rate of achieving a patent airway was comparable between the groups, with a similar occurrence of complications [3]. 

The patient gave written informed consent for the anaesthetic procedure and for publication of this case report, including accompanying images. Effect on adjacent soft tissue are shown in Table 1. The patient was a 55-year-old male, 109 kg in weight and 183 cm in height [Body Mass Index 32.5 kg/m^2^], admitted to our centre to perform a diagnostic neuroradiological procedure. He was classified as American Society of Anesthesiologists status of II with no indications of a difficult airway (El-Ganzouri Risk Index 2, with Mallampati 1, thyromental distance 6.5 cm, inter-incisor distance 4 cm full neck movement, normal upper lip bite test, and no history of previous difficult airway management). Radiological images were obtained to assess the supraglottic use of the Spritztube (distal cuff inflated in oesophagus, proximal cuff inflated in oropharynx). Angiography was performed for diagnostic purposes for suspected aneurysm. The general anaesthesia protocol, intra-operative monitoring, and postoperative care performed for the patient included HR, non-invasive blood pressure, peripheral oxygen saturation, and continuous electrocardiography. The premedication used was midazolam 0.03 mg kg^−1^, followed by fentanyl 2 μg kg^−1^, and propofol 2 mg kg^−1^ for induction of anaesthesia. Anaesthesia was maintained with propofol (continuous infusion 6 to 8 mg kg^−1^ h^−1^) and remifentanil (continuous infusion 0.25 to 0.75 μg kg^−1^ min^−1^). Neither nitrous oxide nor a neuromuscular blocking agent were given, and depth of anaesthesia was monitored with the bispectral index (BIS monitor; Covidien Medical, Louisville, CO, USA), targeting the range 40 to 60. The Spritztube was inserted by trained anaesthesiologist with 10 years of experience. 

The Spritztube was then inserted based on the blind insertion technique after anaesthetic depth. Once the Spritztube was positioned (marker at the level of the incisors) and effective ventilation obtained, we visually assessed the depth of insertion and the impact of the Spritztube on adjacent soft tissues through fluoroscopic images (Figure 5). 

The depth of insertion was assessed: 

(1) Relative to hyoid bone. As the epiglottis is located behind the hyoid bone, the depth of insertion relative to the glottic opening was defined as the perpendicular distance between the hyoid bone and the tip of the Spritztube device in the upper oesophageal sphincter on sagittal images (Figure 5D,F).

(2) Relative to the cervical spine. This distance was defined as the distance between the superior surface of the body of the third cervical vertebra and the tip of the Spritztube on sagittal images (Figure 5D,F).

(3) The position of the hyoid bone relative to the cervical spine. This measurement was determined as the distance between the ventral margin of the hyoid body and the anterior margin of the body of third cervical vertebra (Figure 5).

(4) The position of the ventral margin of the hyoid body relative to the mental spine of the mandible. This measurement was determined by calculating the distance between the medial portion of the hyoid bone and the lower margin of the mandible.

The effect of cuff inflations on tracheal diameter was also assessed.

All detected measurements are shown in Table 1. 

## 3. Discussion and Conclusions

The Spritztube is a new extraglottic device designed to be used in general anaesthesia during spontaneous or positive pressure ventilation, or as alternative to endotracheal intubation, particularly in emergency situations. In this case report, the radiological position of the Spritztube in a patient was evaluated for the first time using fluoroscopic images in addition to clinical assessments. The insertion and depth of the device, placed as endotracheal tube, correct tip positioning, effects of the distal and proximal cuffs on adjacent soft tissues, and the position of the pharyngeal cuff from the cranial to the hyoid bone were evaluated. 

The correct tip positioning could be affected by body weight, oropharyngeal and neck anatomy, and relative positions of the hyoid, mandibular, and other structures which are simultaneously relevant aspects resulting in a difficult or failed intubation [4]. In this case, Body Mass Index suggested moderate obesity but any other parameter influencing airway management was deemed to be normal according to a recently published consensus [5], and no difficulty was recorded or experienced for the Spritztube insertion. The first aim of our report was to assess the relations between the ST and neck anatomy using the fluoroscopy, allowing also possible comparisons with extraglottic devices already in use. Some considerations should be performed. Firstly, the insertion of the Spritztube is mediated by a rigid stylet that is effective when used by experienced operators. In this case, the clinician must be alert to the advancement of the rigid stylet device in the airways as it has a high potential for lacerations and perforations of the oropharyngeal mucosa and subglottic structures. Secondly, although adverse effects due to high intracuff pressures may not commonly cause serious adverse effects, it would be prudent to minimise this risk by ensuring the availability of cuff pressure manometers. 

Thirdly, Spritztube is a re-useable device: not all protein material can be removed by routine cleaning and this raises theoretical concerns over cross-infection risk, hence steam autoclaving is the recommended method of sterilising this device. New Spritztube devices will be soon released on the market for a single use.

The role of extraglottic devices is well recognised for non-operating room anaesthesia, and the novelty of our report is in adding our new device to the portfolio of available alternatives. Particularly, some specific features of our device might offer interesting advantages, such as the possibility of early switching from extraglottic position to endotracheal intubation. In addition, since fluoroscopy is available for the neuroradiologic procedure, our case report also describes the depth and position of the Spritztube relative to soft tissues and the cervical spine. The resulting anterior displacement of the hyoid bone is marginal and different when compared to other extraglottic airway devices, such as i-gel™ and LMA-Supreme™ [6]. 

This specific finding might have important implications, given the evidence that extraglottic device cuff hyper-inflation-related damage to the lingual, hypoglossal, recurrent, and glosso-pharyngeal nerves seems to be a consequence of compression against the hyoid bone [1,7,8]. These findings need to be prudently interpreted taking into account inter-individual anatomical variability of neck anatomy as well as the single case within our report. Although those measurements indicate the effects of Spritztube on the laryngeal structures, magnetic resonance imaging, which would be not really feasible, may better confirm, define the position of the device, and clarify the correlation between these parameters and the anthropological airway anatomy parameters. 

Further studies are necessary to evaluate the safety, efficacy, and potential for the Spritztube in airway management.

## Figures and Tables

**Figure 1 diagnostics-12-02907-f001:**
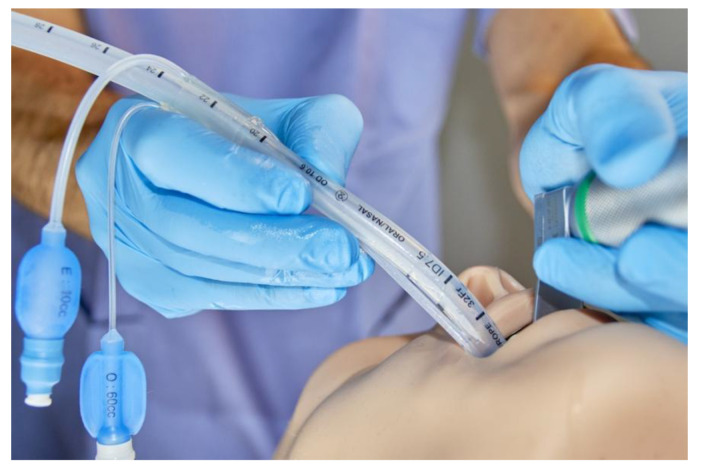
Spritztube details. The device, after insertion into the oral cavity, is stopped once the black marker on the Spritztube (teeth mark) reaches the upper incisors.

**Figure 2 diagnostics-12-02907-f002:**
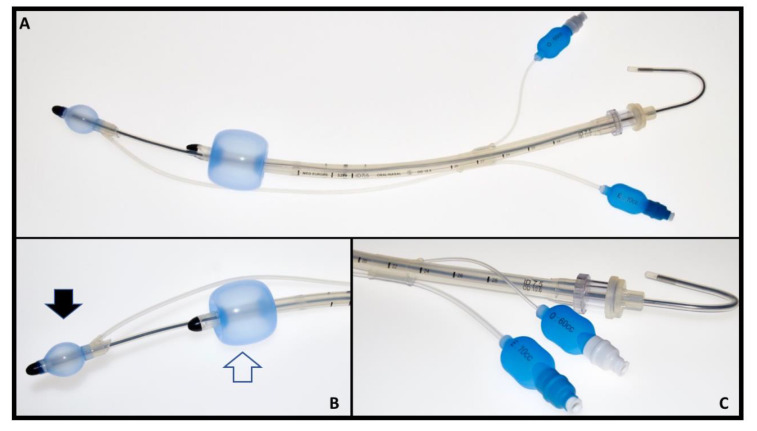
Spritztube characteristics (**A**): device assembled for positioning; note the stylet inside. (**B**): details of distal (oesophageal) cuff (solid arrow) and oropharyngeal cuff (empty arrow). (**C**): colour-coded pilot balloons for oesophageal and oropharyngeal cuffs with respective (10 and 60 cc) inflation volumes.

**Figure 3 diagnostics-12-02907-f003:**
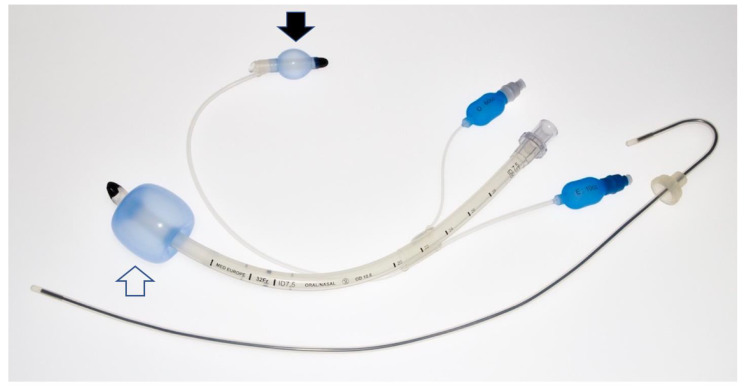
Spritztube details. The device assembled as if in definitive position; note the removed stylet. Once placed as extraglottic device, the distal (oesophageal) cuff (solid arrow) closes the upper oesophageal sphincter while the oropharyngeal cuff (empty arrow) seals the oropharynx. If intubation is performed, the proximal cuff is left inflated and in place, the oropharyngeal cuff deflated and the tube is addressed in trachea with direct view (flexible scope or video-stylet).

**Figure 4 diagnostics-12-02907-f004:**
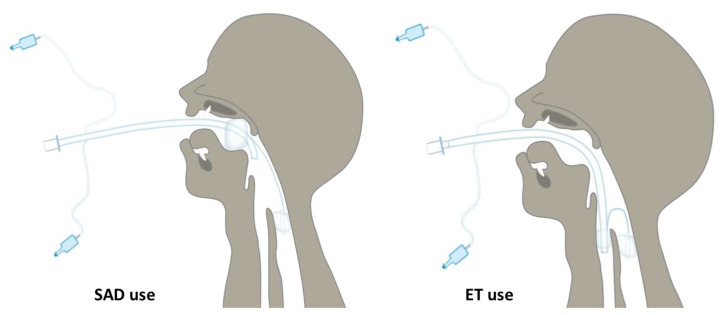
Spritztube in position. Schematic representation of Spritztube in position; on left the extraglottic use (proximal cuff inflated in oropharynx and distal cuff inflated in upper oesophagus). On the right, endotracheal tube use (proximal cuff inflated in trachea, distal cuff left in place in the upper oesophagus).

**Figure 5 diagnostics-12-02907-f005:**
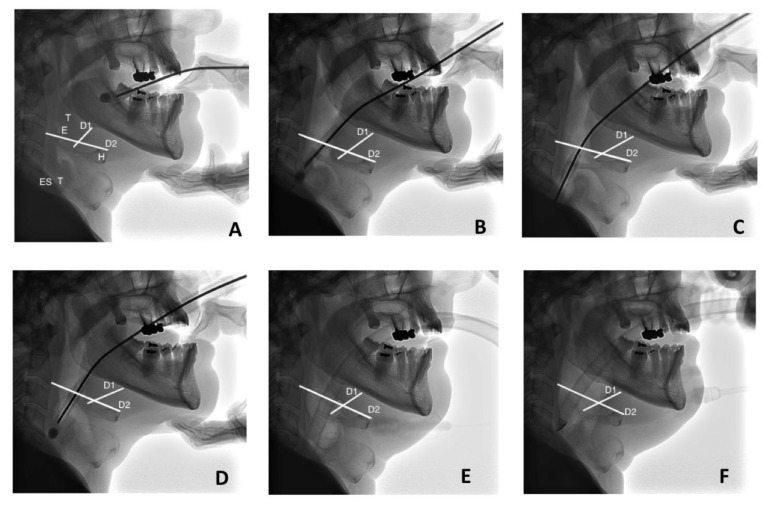
Radiological Localisation of Spritztube (ST) as extraglottic device. Dash lines are used to show the depth of insertion and the effect on trachea diameter. Continuous lines are used to show the effect of the ST on adjacent soft tissues. Held as a pen, the device was inserted into the oral cavity (**A**) following the soft palate (**B**,**C**), stopping introduction once the black marker on the Spritztube (teeth mark) reached the upper incisors (**D**). Then, distal and proximal cuffs were inflated, the stylet removed and the tube connected to the breathing circuit (**E**,**F**). (**A**–**D**): notice the presence of radio-opaque ST stylet, subsequently removed in (**E**,**F**). (**F**) shows oropharyngeal cuff inflated. Abbreviations: D1, Distance 1; D2, Distance 2; E, Epiglottis; Es, Oesophagus; H, Hyoid Bone; T, Trachea.

**Table 1 diagnostics-12-02907-t001:** Effect on adjacent soft tissue after device insertion and with cuffs inflated.

Depth of Insertion Parameters	Measurement (cm)
Cervical vertebra to tip Oesophageal Oropharyngeal	0.87 1
Hyoid bone to tip Oesophageal Oropharyngeal	8.74 8.49
Mandible to tip Oesophageal Oropharyngeal	11.8 8.66
Effect on trachea diameter Oesophageal Oropharyngeal	1.35 2.85
Position of hyoid body relative to mental spine of mandible Oesophageal Oropharyngeal	8.15 7.02
Position of hyoid bone relative to cervicale spine Oesophageal Oropharyngeal	7.40 8.95

## Data Availability

Data sharing is not applicable to this article as no datasets were generated or analysed during the current study.

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
