# Peer review of "Radiologic Imaging of the In Vivo Position of the New Supraglottic Airway Device Spritztube® in an Adult Patient—A Case Report"

_diagnostics, 2022, doi:10.3390/diagnostics12122907_

Round 1

Reviewer 1 Report

The authors present a Case report of a patient who had a Spritztube as airway management device during anesthesia care for a neuroradiological intervention. The Spritztube is a new supraglottic airway device combining the ability of extraglottic ventilation with the possibility to perform tracheal intubation without changing the device.

The Spritztube may be of interest to the anesthesia specialty, as it is a new device, not commercially available and currently under scientific evaluation.

The manuscript is well written, although there is need of substantial revision to clarify the radiological findings.

The following remarks should help the authors to revise the manuscript.

Line 30: SGA do not enable protection oft he airway, but oxygenation and ventilation by creating a more or less stable seal. Please rephrase this sentence.

Line 32: Is the Spritztube a supraglottic or extraglottic device, exspecially since the distal cuff i positioned not above the glottis?

Line 38: Again extraglottic versus supraglottic.

Line 39: Is tracheal intubation possible only with fibreotic assistance or also with direct or indirect laryngoscopy, as you describe in line 55?

Fig 1: You describe in the capture (Line 44) that the oesophageal cuff has a filling volume of 100 cc. The image shows the oesophageal cuff much smaller than the orophayngeal cuff. Please specify the correct filling volumes.
Just a small comment: Part B and C (partly) come with a frame, part A not.

Line 54: Is the stylet preformed or straight? Are the users instructed to preform the stylet? You describe to advance the device until resistance is felt. What is causing resistance?

Line 50: You describe for tracheal intubation the proximal cuff remains inflated. Isn´t that the prospective tracheal cuff? How would you intubate the trachea with a filled cuff?

Line 51: Please replace „adressed“ to „advanced“ and „..under direct view“

Line 55: I do not understand what you mean by „assisted placement with either a SGA or an endotracheal tube“

Line 57: Please refer to the marker for adequate depth of insertion. Is it a „one-size-fits-all device? Is there a range of insertion depth, as with the laryngeal tube? Please show an image oft he marker in place.

Line 58: Do you mean the laryngeal opening?  

Line 60: In the capture of figure 1 you used cc instead of ml. Please use the same unit.

Line 63: Again extraglottic versus supraglottic.

Line 65: If tracheal intubation is performed , should the stylet again be placed inside the device? As the device is made of silicone, in my experience the manoeuvrability of silicon devices contrary to PVC tubes is limited.  

Line 91: Please use „classified“ instead of „adressed“.

Line 95: What do you mean by assessing „both“? Both cuffs? Then you may rephrase this sentence.

Line 103: You may write „blind insertion technique“

Line 103: The term „Scheme 1 may not be part of the capture to fugure 4.

Line 101 to 116:

The figure and the capture as well as the explanation in the next needs major reworking to explain the subject to the readers. For example the abbreviations in the image („E, T, D1, D2, ES, T) are not specified. The abbreviation ST in the capture of figure 4 is not specified. Moreover, what is meant by the „tip of the Spritztube“? Is it the distal tip wich should be positioned in the upper oesophageal sphincter? Ist the stylet still in place? Wich cuff was inflated to what extent in what image?
I would recommend a step-by-step approach with image and a corresponding schematic drawing.

Line 118: Table 1 may be rephrased to „Radiological imaging….“

Line 118: Table 1: What is meant by „First Cuff“, what by Second Cuff?“ Please use the same term as above (oesophgeal, oropharyngeal cuff). Please change „intertion“ to „insertion“.

Line 123: This is a radiological position, not an anatomical.

Line 128: Correct tip positioning may indeed be the key factor of extraglottic airway devices. The spritztube consists oft wo tips. Please refer to that in your discussion

Line 131: Do you mean intubation performed as tracheal intubation? Or SGA placement?

Line 134-136: Your report undoubtly show that you placed a Spritztube in a patient undergoing a radiographic procedure, but your conclusion to underline efficacy and safety oft he new device with a single patient case is way to far. Please remove that sentence.

Line 138: Please explain the abbreviations NORA and MAC to the readers.

Line 140-141: As I appreciate the possibility to easyly switch to tracheal intubation without changing the device is attractive I wonder how it works. Is there a case report or an observation in vivo to underline this possibility?  

Line 144: „What do you mean by verisimilarly? Please explain this comparison to other SGA?

Line 147: There is no reference 8

General points:

In my opinion to advance a stiff styleted device into the airway has a high potential for lacerations and perforations of oropharyngeal muscosa and subglottic structures. Please refere to that in your discussion.

How is your experience during removing of the stylet? Does the distal part of the device remains still in place?

An inflation of up to 60 cc may exerts substantial mucosal pressure and subsequent necrosis. Why don´t you recommend a pressure measurement?

What do you think of re-useable airway devices in times of infection prevention control and possible transmissionable infectious deseases ? Please refere to that in your discussion.

In my personal conclusion, MRI images may be much more instructive as for the position of the device. What do you think?  

Author Response

Reviewer 1:

  • The authors present a Case report of a patient who had a Spritztube as airway management device during anesthesia care for a neuroradiological intervention. The Spritztube is a new extraglottic airway device combining the ability of extraglottic ventilation with the possibility to perform tracheal intubation without changing the device. The Spritztube may be of interest to the anesthesia specialty, as it is a new device, not commercially available and currently under scientific evaluation. The manuscript is well written, although there is need of substantial revision to clarify the radiological findings.

Response:  We thank a lot for the comment. Recently, the Spritztube is commercially available so we correct the sentence in the manuscript.  

  • Line 30: extraglottic do not enable protection oft he airway, but oxygenation and ventilation by creating a more or less stable seal. Please rephrase this sentence.

Response: Fully Agree, thanks for the comment. We re-formulated the phrase according to reviewer’s suggestion. (pg2, line 32-33)

  • Line 32: Is the Spritztube a extraglottic or extraglottic device, exspecially since the distal cuff i positioned not above the glottis?

Response: Fully Agree, thanks for the comment. We re-placed extraglottic instead of extraglottic according to reviewer’s suggestion. (pg2, line 34-35)

  • Line 38: Again extraglottic versus extraglottic

Response: Fully Agree, thanks for the comment.  We re-placed extraglottic instead of extraglottic according to reviewer’s suggestion. (pg2, line 34-35)

  • Line 39: Is tracheal intubation possible only with fibreotic assistance or also with direct or indirect laryngoscopy, as you describe in line 55?

    Response: Fully Agree, thanks for the comment. Tracheal intubation is also possible  with direct or indirect laryngoscopy. We added it to better clarify. (pg 2, line 42-44)

  • Fig 1: You describe in the capture (Line 44) that the oesophageal cuff has a filling volume of 100 cc. The image shows the oesophageal cuff much smaller than the orophayngeal cuff. Please specify the correct filling volumes.

Response: Fully Agree, thanks for the comment. It was a mistake. We corrected in the right way. (pg 2, line 49-50)

  • Fig 1: You describe in the capture (Line 44) that the oesophageal cuff has a filling volume of 100 cc. The image shows the oesophageal cuff much smaller than the orophayngeal cuff. Please specify the correct filling volumes. Just a small comment: Part B and C (partly) come with a frame, part A not.

Response: Fully Agree, thanks for the comment. The volume written was wrong and we corrected in the right way. We fixed the figure completing the frame

  • Line 54: Isthe stylet preformed or straight? Are the users instructed to preform the stylet? You describe to advance the device until resistance is felt. What is causing resistance?

Response: Fully Agree, thanks for the comment. Honestly, this is a mistake describing the technique. Held as a pen, the device is inserted into the oral cavity following the soft palate, stopping introduction once the black marker on the Spritztube (teeth mark) reached the upper incisors. We better specified it. Again, thanks a lot.

  • Line 50: You describe for tracheal intubation the proximal cuff remains inflated. Isn´t that the prospective tracheal cuff? How would you intubate the trachea with a filled cuff?

Response: Thanks a lot for the comment. The part that the reviewer’s mentioned was related to extraglottic device positioning. In addition, we decided to remove the following sentence because much confusing :”Then, a direct or indirect laryngoscope can be used to assist placement, with either a extraglottic device or an endotracheal tube.”

  • Line 51: Please replace „adressed“ to „advanced“ and „..under direct view“

Response: Thanks a lot for the comment. The part that the reviewer’s mentioned. We made corrections according to reviewer’s comment (line 67-69).

  • Line 55: I do not understand what you mean by „assisted placement with either a extraglottic or an endotracheal tube“

Response:Thanks a lot for the comment. Fully agree. As mentioned before, we decided to remove this sentece because much confusing.

  • Line 57: Please refer to the marker for adequate depth of insertion. Is it a „one-size-fits-all device? Is there a range of insertion depth, as with the laryngeal tube? Please show an image of the marker in place.

Response: Thanks a lot for the comment. Fully agree. It is one size fits all device. If we understood well,  just in the figure 1 and figure 2 are visible markers. We refer in the text to those two figures. As a normal endotracheal tube, average distances such as the 20/22 or 21/23 (cm, female/male) rule have instead been simply applied. (pg 3, line     70-72).

  • Line 58: Do you mean the laryngeal opening?

Response:Thanks a lot for the comment. Fully agree. We added it in the manuscript. ( pg 3, line 62)

  • Line 60: In the capture of figure 1 you used cc instead of ml. Please use the same unit.

Response:Thanks a lot for the comment. Fully agree. We corrected according to reviewer’s suggestion.

  • Line 63: Again extraglottic versus extraglottic.

Response: Thanks a lot for the comment. Fully agree. We corrected according to reviewer’s suggestion.

  • Line 65: If tracheal intubation is performed , should the stylet again be placed inside the device? As the device is made of silicone, in my experience the manoeuvrability of silicon devices contrary to PVC tubes is limited.

Response: Thanks a lot for the comment. Fully agree. However, if tracheal intubation is performed after extraglottic positioning, the stylet should not be replaced inside the device    (pg 4, line 70-71)

  • Line 91: Please use „classified“ instead of „adressed“.

Response: Thanks a lot for the comment. Fully agree. We corrected according to reviewer’s suggestion. (pg 5, line 99)

  • Line 95: What do you mean by assessing „both“? Both cuffs? Then you may rephrase this sentence.

Response:Thanks a lot for the comment. Fully agree. It was a typo error. We remove the word both (pg 5, line 104)

  • Line 103: You may write „blind insertion technique“

Response:Thanks a lot for the comment. Fully agree. We corrected according to reviewer’s  suggestion. (pg 5, line 105-106)

  • Line 103: The term „Scheme 1 may not be part of the capture to fugure 4.

Response: Thanks a lot for the comment. Fully agree. We corrected according to reviewer’s  suggestion. (pg 5, line 113)

  • Line 101 to 116:The figure and the capture as well as the explanation in the next needs major reworking to explain the subject to the readers. For example the abbreviations in the image („E, T, D1, D2, ES, T) are not specified. The abbreviation ST in the capture of figure 4 is not specified. Moreover, what is meant by the „tip of the Spritztube“? Is it the distal tip wich should be positioned in the upper oesophageal sphincter? Ist the stylet still in place? Wich cuff was inflated to what extent in what image?

I would recommend a step-by-step approach with image and a corresponding schematic drawing.

Response: Thanks a lot for the comment. Fully agree. We corrected according to reviewer’s suggestion. (pg 5, line 113-121)

  • Line 118: Table 1 may be rephrased to „Radiological imaging….“

Response: Thanks a lot for the comment. Fully agree. We corrected according to reviewer’s suggestion.

  • Line 118: Table 1: What is meant by „First Cuff“, what by Second Cuff?“ Please use the same term as above (oesophgeal, oropharyngeal cuff). Please change „intertion“ to „insertion“.

Response: Thanks a lot for the comment. Fully agree. We corrected according to reviewer’s suggestion.

  • Line 123: This is a radiological position, not an anatomical.

Response: Thanks a lot for the comment. Fully agree. We corrected according to reviewer’s suggestion.

  • Line 128: Correct tip positioning may indeed be the key factor of extraglottic airway devices. The spritztube consists oftwo tips. Please refer to that in your discussion

Response: Thanks a lot for the comment. Fully agree. We corrected according to reviewer’s suggestion

  • Line 131: Do you mean intubation performed as tracheal intubation? Or SGA placement?

Response: Thanks a lot for the comment. Fully agree. We corrected according adding that we assessed as endotracheal intubation.

  • Line 134-136: Your report undoubtly show that you placed a Spritztube in a patient undergoing a radiographic procedure, but your conclusion to underline efficacy and safety oft he new device with a single patient case is way to far. Please remove that sentence.

Response:Thanks a lot for the comment.  Fully agree. We remove the sentence as reviewer’s suggestion

  • Line 138: Please explain the abbreviations NORA and MAC to the readers.

Response: Thanks a lot for the comment.  We remove MAC acronym and we just expanded NORA acronym as Non Operating Room Anesthesia

  • Line 140-141: As I appreciate the possibility to easyly switch to tracheal intubation without changing the device is attractive I wonder how it works. Is there a case report or an observation in vivo to underline this possibility?  

Response: Thanks a lot for the comment. This is a case report . I better specified in the manuscript   

  • Line 144: „What do you mean by verisimilarly? Please explain this comparison to other SGA?

Response:Thanks a lot for the comment. We specified in the manuscript the comparison with other devices.

  • Line 147: There is no reference 8

Response:Thanks a lot for the comment. We have reference 8 because we added another reference to specify compraison between devices

  • General points: In my opinion to advance a stiff styleted device into the airway has a high potential for lacerations and perforations of oropharyngeal muscosa and subglottic structures. Please refere to that in your discussion.

Response:Thanks a lot for the comment.  We added this part in the discussion section.

  • How is your experience during removing of the stylet? Does the distal part of the device remains still in place?

Response: Thanks a lot for the comment. This observation is very useful. In fact, once positioned as an endotracheal tube, we recommend that you capsize the distal part of the tube which can remain in place or you can take it out of the mouth and cut it. If the reviewer sees fit, we can add this part to the discussion.

  • An inflation of up to 60 cc may exerts substantial mucosal pressure and subsequent necrosis. Why don´t you recommend a pressure measurement?

Response: Thanks a lot for the comment. This observation is very useful. We added it in the manuscript according to reviewer’s suggestion.

  • What do you think of re-useable airway devices in times of infection prevention control and possible transmissionable infectious deseases ? Please refere to that in your discussion.

 Response:Thanks a lot for the comment. Fully agree.  We added this part in the manuscript: “Spritztube is a re-useble device: not all protein material can be removed by routine cleaning and this raises theoretical concerns over cross-infection risk, hence steam autoclaving is the recommended method of sterilising this device. New spritztube will be sooner release on market for single use”.

  • In my personal conclusion, MRI images may be much more instructive as for the position of the device. What do you think?

Response: Thanks a lot for the comment. Fully agree. We corrected according to reviewer’s suggestion

Reviewer 2 Report

MS Title: Radiologic imaging of the in vivo position of the new supra-glottic airway device Spritztube ® in an adult patient. A case report.

MS authors: Silvia DE ROSA et al.

MS No: diagnostics-1942877

This is a short case report of using Spritztube® (a new supraglottic airway device) in a patient receiving neuro-radiological examination. Radiologic imaging of the Spritztube (ST) in situ provided the assessment of the depth of insertion of such airway device and its relationship to the anatomical structures in vivo. Several parameters were studied and the results suggested such Spritztube could be helpful in maintaining an adequate airway during radiologic procedures.

Comments as follows:

Page 2, Figure 1, Line 44: 100 ml or 20 ml for distal cuff?

Page 3, Figure 3:

l   SAD use: When 60 ml of the proximal cuff inflated in the oropharynx area, how to ensure the seal pressure and preventive effect on leak? What is the rationale of such 60 ml cuff contour incorporating into different oropharyngeal anatomy?

l   ET use: When Spritztube was secured into trachea functioned as a endotracheal tube, what is the need to inflate the distal cuff in the esophagus? In addition, in this case, will the 60-ml design of the proximal cuff inside the trachea be feasible? What is the design rationale of such 60 ml cuff balloon (e.g., high volume/low pressure? shape of the cuff)?

Page 3, line 55 and line 64: What is the need to use DL or VL to assist placement in the presumed common practice of Spritztube, if the main intent of ST is used in a blind manner?

Page 4, case presentation: The peri-anesthesia monitoring and medication regimen should be briefly mentioned. Or refer to the relevant literature from the same research group. In addition, the clinical diagnosis and purpose of this neurological exam should be mentioned.

Page 4, line 85: “A patient enrolled in a first-in-human trial of the Spritztube  vs. an Laryngeal Mask Airway Supreme™ in anaesthetised adult patients (NCT03443219)” The author should clarify the relationship between this case report and aforementioned clinical study. “they gave written informed consent for the anaesthetic procedure and for publication of this case report,..” should be “this man gave written IC….”.   

Page 4, Line 102, Figure 4: The abbreviation of ST (Spritztube?); Scheme 1? The labels (D1,D2, E, T, H? The labels and interpretlation of each panel? It will be helpful to explain each panel to assist the readers to easily catch the points of the Figure 4 (especially the anatomical relation with the ST).

Page 4, line 107-117: “1)” is missing. Also, it would be nice to correlate the information from this paragraph with those in the Figure 4.

Page 5, Table 1: some typo-errors: “intertion” “manibole”. Again, it would be nice if graphic illustration on all the presenting parameters can be made to facilitate the understanding of the meaning of the parameters. Also, the meaning and significance of all the numbers of the measurement (cm) in the Table 1 need to be discussed. For example, how the possible correlation between these parameters and the anthropological airway anatomy parameters.

Page 5, line 128-138: About the discussion, the authors, based on this single case report, made a conclusion that ST‘s safety and effectiveness are convincingly acceptable. The authors need to define and quantify the outcome parameters of “safety and effectiveness” before reaching that kind of conclusion.

Page 5, line 135 and page 6, line 141 and the last sentence of the Summary: The authors conclude from this case report as follows: “Our report shows that Spritztube could be helpful in maintaining an adequate airway during radiologic procedures”. However, from this case report, it seems the fluoroscopy was purposely used to assess the relations between the ST and neck anatomy.    

Regarding one article the authors published earlier. Certain similar figures have already appeared in this article. Any reason not to mention this reference in the present manuscript?

De Rosa S, Messina A, Sorbello M, Rigobello A, Colombo D, Piccolo A, Bonaldi E, Gennaro P, Urukalo V, Pellizzari A, Bonato R, Carboni SC. Laryngeal Mask Airway Supreme vs. the Spritztube tracheal cannula in anaesthetised adult patients: A randomised controlled trial. Eur J Anaesthesiol. 2019 Dec;36(12):955-962. doi: 10.1097/EJA.0000000000001106.

Author Response

Reviewer 2:

This is a short case report of using Spritztube® (a new supraglottic airway device) in a patient receiving neuro-radiological examination. Radiologic imaging of the Spritztube (ST) in situ provided the assessment of the depth of insertion of such airway device and its relationship to the anatomical structures in vivo. Several parameters were studied and the results suggested such Spritztube could be helpful in maintaining an adequate airway during radiologic procedures.

Response:We really thank the reviewer for this kind comment.

Comments as follows:

  • Page 2, Figure 1, Line 44: 100 ml or 20 ml for distal cuff?

Response: Thanks for the comment. The correct value is 10 ml.

  • Page 3, Figure 3:
    • SAD use: When 60 ml of the proximal cuff inflated in the oropharynx area, how to ensure the seal pressure and preventive effect on leak? What is the rationale of such 60 ml cuff contour incorporating into different oropharyngeal anatomy?

Response:Thanks for the comment. The cuff is made on silicon with low compliance, and 60 ml are enough to maintain Pcuff t 20–30 cm H2O with adequate tension. However, this should be confirmed in vitro analysis to improve sealing properties of tracheal tubes.

  • ET use: When Spritztube was secured into trachea functioned as a endotracheal tube, what is the need to inflate the distal cuff in the esophagus? In addition, in this case, will the 60-ml design of the proximal cuff inside the trachea be feasible? What is the design rationale of such 60 ml cuff balloon (e.g., high volume/low pressure? shape of the cuff)?

Response: Thanks for the comment.  The need to inflate distal cuff is to avoid aspiration.Again, the design of this cuff is related to silicon material that has low compliance.

  • Page 3, line 55 and line 64: What is the need to use DL or VL to assist placement in the presumed common practice of Spritztube, if the main intent of ST is used in a blind manner?

 Response: Thanks a lot for the comment. The ST is designed not only as extraglottic device but also to perform endotracheal intubation if it is required during anesthetic management or in case of difficult intubation. In the latter case after extraglottic ventilation, it could be positioned as endotracheal tube.

Page 4, case presentation: The peri-anesthesia monitoring and medication regimen should be briefly mentioned. Or refer to the relevant literature from the same research group. In addition, the clinical diagnosis and purpose of this neurological exam should be mentioned.

Response: Thanks for the comment. Angiography was performed for diagnostic purposes for suspected aneurysm.  The general anaesthesia protocol, intra-operative monitoring and postoperative care performed for the patient included HR, peripheral oxygen saturation and continuous electrocardiography. The premedication  perfromed was with midazolam 0.03 mg kg−1 followed by fentanyl 2 μg kg−1 and propofol 2 mg kg−1 for induction of anaesthesia. Anaesthesia was maintained with propofol (continuous infusion 6 to 8 mg kg−1 h−1) and remifentanil (continuous infusion 0.25 to 0.75 μg kg−1 min−1). Neither nitrous oxide nor a neuromuscular blocking agent were given, and depth of anaesthesia was monitored with the bispectral index (BIS monitor; Covidien Medical, Louisville, Colorado, USA), targeting the range 40 to 60. The Spritztube were inserted by trained anaesthesiologist with 10 years of experience.  

We added it in the manuscript.

Page 4, line 85: “A patient enrolled in a first-in-human trial of the Spritztube  vs. an Laryngeal Mask Airway Supreme™ in anaesthetised adult patients (NCT03443219)” The author should clarify the relationship between this case report and aforementioned clinical study. “they gave written informed consent for the anaesthetic procedure and for publication of this case report,..” should be “this man gave written IC….”.

Response: Thanks for the comment. We made corrections according to reviewer’s suggestions.

Page 4, Line 102, Figure 4: The abbreviation of ST (Spritztube?); Scheme 1? The labels (D1,D2, E, T, H? The labels and interpretlation of each panel? It will be helpful to explain each panel to assist the readers to easily catch the points of the Figure 4 (especially the anatomical relation with the ST).

Response: Thanks for the comment. We made corrections according to reviewer’s suggestions.

Page 4, line 107-117: “1)” is missing. Also, it would be nice to correlate the information from this paragraph with those in the Figure 4.

Response: Thanks for the comment. We made corrections according to reviewer’s suggestions.

Page 5, Table 1: some typo-errors: “intertion” “manibole”. Again, it would be nice if graphic illustration on all the presenting parameters can be made to facilitate the understanding of the meaning of the parameters. Also, the meaning and significance of all the numbers of the measurement (cm) in the Table 1 need to be discussed. For example, how the possible correlation between these parameters and the anthropological airway anatomy parameters.

Response:: Thanks for the comment. We made corrections according to reviewer’s suggestions. However, those measurements indicate the effects of Spritztube on the laryngeal structures. Illustration is to difficult to perform and MRI will better clarify the correlation between these parameters and the anthropological airway anatomy parameters. We added at the end of the discussion

Page 5, line 128-138: About the discussion, the authors, based on this single case report, made a conclusion that ST‘s safety and effectiveness are convincingly acceptable. The authors need to define and quantify the outcome parameters of “safety and effectiveness” before reaching that kind of conclusion.

 Response: Thanks for the comment. We removed this phrase from the manuscript because uncorrect information as reviewer indicated We changed the phrase replacing with “The first aim of our report was  to assess the relations between the ST and neck anatomy using the fluoroscopy”.

Page 5, line 135 and page 6, line 141 and the last sentence of the Summary: The authors conclude from this case report as follows: “Our report shows that Spritztube could be helpful in maintaining an adequate airway during radiologic procedures”. However, from this case report, it seems the fluoroscopy was purposely used to assess the relations between the ST and neck anatomy.    

 Response: Thanks for the comment. We removed this phrase from the manuscript because uncorrect information as reviewer indicated

Regarding one article the authors published earlier. Certain similar figures have already appeared in this article. Any reason not to mention this reference in the present manuscript?

De Rosa S, Messina A, Sorbello M, Rigobello A, Colombo D, Piccolo A, Bonaldi E, Gennaro P, Urukalo V, Pellizzari A, Bonato R, Carboni SC. Laryngeal Mask Airway Supreme vs. the Spritztube tracheal cannula in anaesthetised adult patients: A randomised controlled trial. Eur J Anaesthesiol. 2019 Dec;36(12):955-962. doi: 10.1097/EJA.0000000000001106.

Response:: Thanks for the comment. We added the publications in bibliography and we referenced in the manuscript.

Round 2

Reviewer 1 Report

Dear authors,

thank you for the revision of your manuscript, wich in my opinion is much more precise than the first draft but still needs some reworking.

Please consider the following remarks as they may help to bring your manuscript to a publishable standard.

Line 31: I would omit "stable"

Line 41: Change "Stilette" to "stylet"

Line 61: "is inserted" instaed of "was inserted"

Line 63: teeth marker: are they visible on the Figure 2? It may be helpful to the readers if this markers are not only described in the text but shown in the figure

Line 69: Where exactly you recommend a cuff pressure manometer to limit the muscosal pressure?

Line 87: Please change "experimental study" to "case report"

Line 96: Please specify that the patient was part of a clinical trial published earlier and reference the study.

Line 105/106: This sentence makes no sense to me, sorry. Table 1 contains a complete different topic

Line 109: ASA KLassifikation is a "Status", not a "Score"

Line 114: You write a "diagnostic angiography", later you write "interventional angiography". Please clarify what type of procedure/intervention the patient had.

Line 114-116: Was the blood pressure not measured during the angiography?

Line 125-127: it sounds a bit akward in a new device to call it "standard technique; moreover, if you have a BIS-monitor in place, why rely on the response to the jaw thrust?

Capture Fig 4: omit "blind insertion technique"

Table 1: I am sorry, but th einformation given in Table 1 is still not clear to me. The headline of the Table is promising "Airway evaluation, depth of insertion and effect on adjacent soft tissue. I can only find effects on adjacent soft tissue.

Please tell the reader what you want to show with the table, take an informative headline for the table, and consider rephrasing the table column called "depth of insertion parameters" . I don´t get the point for this column: is the cuff already inflates? Why "on insertion"?
Change "mandibol" to "mandible"

Line 162: Extraglotticinstead of SGA

Line 164: Again: this is a radiological/fluoroscopic position, not an anatomical.

Line 207: MRI to confirm the position would be not really feasible, but to characterise the position of the ST MRI  may be helpful

Line 214: List of Abbreviations:
I would add  ST: Spritztube
SGA:

Author Response

Reviewer 1:

Dear authors,

thank you for the revision of your manuscript, wich in my opinion is much more precise than the first draft but still needs some reworking.

Please consider the following remarks as they may help to bring your manuscript to a publishable standard.

Response:: Thanks for the comment.  We greatly appreciate your comment!

Line 31: I would omit "stable"

Response: Fully agree. We removed “stable” according to reviewer’s suggestion

Line 41: Change "Stilette" to "stylet"

Response: Fully agree. We modified it according to reviewer’s suggestion

Line 61: "is inserted" instaed of "was inserted"

Response: Fully agree. We modified it according to reviewer’s suggestion

Line 63: teeth marker: are they visible on the Figure 2? It may be helpful to the readers if this markers are not only described in the text but shown in the figure

Response: Fully agree. We added a figure and reformatted the figures it according to reviewer’s suggestion

Line 69: Where exactly you recommend a cuff pressure manometer to limit the muscosal pressure?

Line 87: Please change "experimental study" to "case report"

Response: Fully agree. We modified it according to reviewer’s suggestion

Line 96: Please specify that the patient was part of a clinical trial published earlier and reference the study.

Response: Fully agree. We modified it according to reviewer’s suggestion

Line 105/106: This sentence makes no sense to me, sorry. Table 1 contains a complete different topic

Response: Fully agree. We modified it according to reviewer’s suggestion

Line 109: ASA KLassifikation is a "Status", not a "Score"

Response: Fully agree. We modified it according to reviewer’s suggestion

Line 114: You write a "diagnostic angiography", later you write "interventional angiography". Please clarify what type of procedure/intervention the patient had.

Response: Fully agree. We modified it according to reviewer’s suggestion

Line 114-116: Was the blood pressure not measured during the angiography?

Response: Fully agree. Yes it is. We modified it according to reviewer’s suggestion

Line 125-127: it sounds a bit akward in a new device to call it "standard technique; moreover, if you have a BIS-monitor in place, why rely on the response to the jaw thrust?

Response: Fully agree. We just leave the phrase in that way: “The Spritztube was then inserted based on blind insertion technique after anaesthetic depth”

Capture Fig 4: omit "blind insertion technique"

Response: Fully agree. Yes it is. We modified it according to reviewer’s suggestion

Table 1: I am sorry, but th einformation given in Table 1 is still not clear to me. The headline of the Table is promising "Airway evaluation, depth of insertion and effect on adjacent soft tissue. I can only find effects on adjacent soft tissue.

Response: Fully agree. Yes it is. We modified it according to reviewer’s suggestion

Please tell the reader what you want to show with the table, take an informative headline for the table, and consider rephrasing the table column called "depth of insertion parameters" . I don´t get the point for this column: is the cuff already inflates? Why "on insertion"?

Response: Fully agree. Yes it is. We modified it according to reviewer’s suggestion .

Change "mandibol" to "mandible"

Response: Fully agree. Yes it is. We modified it according to reviewer’s suggestion .

Line 162: Extraglotticinstead of SGA

Response: Fully agree. Yes it is. We modified it according to reviewer’s suggestion .

Line 164: Again: this is a radiological/fluoroscopic position, not an anatomical.

Response: Fully agree. Yes it is. We modified it according to reviewer’s suggestion .

Line 207: MRI to confirm the position would be not really feasible, but to characterise the position of the ST MRI  may be helpful

Response: Fully agree. Yes it is. We modified it according to reviewer’s suggestion .

Line 214: List of Abbreviations:

I would add  ST: Spritztube

SGA:

Response: Fully agree. Yes it is. We modified it according to reviewer’s suggestion . We removed SGA abbreviation

Response:: Thanks for the comment.  

Reviewer 2 Report

Dear Dr. De Rosa et al.

Thanks for your response and revision. I have no further questions. Congratulations for your research contribution to the academic communities. 

best

Author Response

Dear Dr. De Rosa et al.

Thanks for your response and revision. I have no further questions. Congratulations for your research contribution to the academic communities. 

best

Response: Thanks a lot for your kind comment